# The Ancient and Evolved Mouse Sperm-Associated Antigen 6 Genes Have Different Biologic Functions In Vivo

**DOI:** 10.3390/cells11030336

**Published:** 2022-01-20

**Authors:** Yi Tian Yap, Wei Li, Qi Zhou, Sarah Haj-Diab, Dipanwita Dutta Chowdhury, Asmita Vaishnav, Pamela Harding, David C. Williams, Brian F. Edwards, Jerome F. Strauss, Zhibing Zhang

**Affiliations:** 1Department of Physiology, School of Medicine, Wayne State University, Detroit, MI 48201, USA; gq5215@wayne.edu (Y.T.Y.); waylee@wayne.edu (W.L.); hc6441@wayne.edu (Q.Z.); sarah.haj-diab@wayne.edu (S.H.-D.); 2Department of Occupational and Environmental Medicine, School of Public Health, Wuhan University of Science and Technology, Wuhan 430060, China; 3Department of Biochemistry, Microbiology & Immunology, School of Medicine, Wayne State University, Detroit, MI 48201, USA; dipanwita160291@gmail.com (D.D.C.); vaishnav@med.wayne.edu (A.V.); brian.edwards@wayne.edu (B.F.E.); 4Hypertension & Vascular Research Division, Henry Ford Health System, Detroit, MI 48201, USA; pharding@wayne.edu; 5Department of Pathology and Laboratory Medicine, University of North Carolina at Chapel Hill, Chapel Hill, NC 27599, USA; david_willjr@med.unc.edu; 6Center for Research on Reproduction and Women’s Health, Department of Obstetrics and Gynecology, Perelman School of Medicine, University of Pennsylvania, Philadelphia, PA 19104, USA; jerome.strauss@vcuhealth.org; 7The C.S. Mott Center for Human Growth and Development, Department of Obstetrics & Gynecology, Wayne State University, Detroit, MI 48201, USA

**Keywords:** cilia, flagella, axoneme, central apparatus, spermatogenesis, *Spag6*, *Spag6l*

## Abstract

Sperm-associated antigen 6 (SPAG6) is the mammalian orthologue of *Chlamydomonas PF16*, an axonemal central pair protein involved in flagellar motility. In mice, two *Spag6* genes have been identified. The ancestral gene, on mouse chromosome 2, is named *Spag6*. A related gene originally called *Spag6, localized* on mouse chromosome 16, evolved from the ancient *Spag6* gene. It has been renamed *Spag6-like* (*Spag6l*). *Spag6* encodes a 1.6 kb transcript consisting of 11 exons, while *Spag6l* encodes a 2.4 kb transcript which contains an additional non-coding exon in the 3′-end as well as the 11 exons found in *Spag6*. The two *Spag6* genes share high similarities in their nucleotide and amino acid sequences. Unlike *Spag6l* mRNA, which is widely expressed, *Spag6* mRNA expression is limited to a smaller number of tissues, including the testis and brain. In transfected mammalian cells, SPAG6/GFP is localized on microtubules, a similar localization as SPAG6L. A global *Spag6l* knockout mouse model was generated previously. In addition to a role in modulating the ciliary beat, SPAG6L has many unexpected functions, including roles in the regulation of ciliogenesis/spermatogenesis, hearing, and the immunological synapse, among others. To investigate the role of the ancient *Spag6* gene, we phenotyped global *Spag6* knockout mice. All homozygous mutant mice were grossly normal, and fertility was not affected in both males and females. The homozygous males had normal sperm parameters, including sperm number, motility, and morphology. Examination of testis histology revealed normal spermatogenesis. Testicular protein expression levels of selected SPAG6L binding partners, including SPAG16L, were not changed in the *Spag6* knockout mice, even though the SPAG16L level was significantly reduced in the *Spag6l* knockout mice. Structural analysis of the two SPAG6 proteins shows that both adopt very similar folds, with differences in a few amino acids, many of which are solvent-exposed. These differences endow the two proteins with different functional characteristics, even though both have eight armadillo repeats that mediate protein–protein interaction. Our studies suggest that SPAG6 and SPAG6L have different functions in vivo, with the evolved SPAG6L protein being more important. Since the two proteins have some overlapping binding partners, SPAG6 could have functions that are yet to be identified.

## 1. Introduction

*Chlamydomonas P**F16* encodes a protein localized in the central pair of the axoneme that modulates flagellar motility [1]. Its mammalian orthologue, sperm-associated antigen 6 (*Spag6*), was first cloned by screening a human cDNA library using serum from an infertile man who had a high titer of anti-sperm antibodies in his blood [2]. The mouse *Spag6* gene was subsequently cloned by screening a mouse testis cDNA library using a probe that was generated by a polymerase chain reaction from a mouse EST clone that had a high homology to human *SPAG6* [3]. The mouse *Spag6* gene was mapped to chromosome 16 using the T31 radiation hybrid panel [3]. Human and mouse SPAG6 proteins are abundant in sperm tails. In transfected mammalian cells, SPAG6 decorates the microtubule [3].

To study the function of the mammalian *Spag6* gene, a global *Spag6* knockout mouse model was generated. The homozygous *Spag6* mutant mice were significantly impaired, with 50% of mutant mice exhibiting premature death attributed to the development of hydrocephalus. In the male mice that survived to adulthood, infertility was a prominent phenotype, arising from a poor sperm motility and defective flagellar organization [4]. In addition to the modulation of cilia/sperm motility, other functions of the mouse *Spag6* gene were subsequently discovered. In the absence of SPAG6, motile ciliogenesis, axoneme orientation, and polarity were altered in the trachea and middle ear epithelial cells [5,6]. Mouse SPAG6 also regulates fibroblast cell growth, morphology, migration, and primary ciliogenesis [7]. Immune synapse formation/function as well as spiral ganglion neuron development were also affected in the absence of SPAG6 [8,9]. More recent studies demonstrated that mouse SPAG6 is required for hearing and spermatogenesis [10,11,12,13]. These studies demonstrate that mouse SPAG6 has broader functions than cilia/flagella motility.

PF16/SPAG6 proteins have eight armadillo repeats that mediate protein–protein interaction [1]. With the aid of yeast two-hybrid screening, it was discovered that SPAG6 is associated with a number of binding partners with different functions [10]. Diverse functions for SPAG6 have also been identified in other species, including fish, chickens, and pigs [14,15,16,17,18,19]. In humans, mutations in the *SPAG6* gene have been associated with severe asthenoteratospermia, characterized by multiple flagellar malformations, resulting in sterility [20]. In addition, studies have shown that SPAG6 expression is significantly increased in a number of cancers [21,22,23,24,25,26,27,28], which further supports the notion that mammalian SPAG6 performs other functions in addition to modulating cilia/flagella motility.

Recently, another mouse *Spag6* gene was discovered. It was originally named *Spag6-like (Spag6l)* or *Spag6*-*BC061194* [29]. The *Spag6l* gene, localized on chromosome 2, is in fact the parental gene of *Spag6*, and the *Spag6* gene localized on chromosome 16 was derived from a duplication of the parental gene during evolution as analyzed by a phylogenetic tree analysis [29]. Therefore, the parental *Spag6l* gene was renamed *Spag6*, and the evolved *Spag6* gene is now named *Spag6l*.

In this study, we characterized the mouse *Spag6* gene and phenotyped global *Spag6* knockout (KO) mice. Although the two *Spag6* genes share high similarities in their nucleotide and amino acid sequences, the few different amino acids alter their functions. Surprisingly, mice lacking SPAG6 did not show any gross abnormalities. Hydrocephalus was not discovered in any of the mice analyzed. All homozygous mutant mice examined were fertile, and the males showed normal spermatogenesis and sperm parameters. Testicular expression levels of selected proteins that were down-regulated in *Spag6l* KO mice were not changed in the *Spag6* KO mice. This phenotype was totally different from that of the *Spag6l* KO mice, indicating that the ancient mouse *Spag6* gene may have lost key functions during evolution. Conversely, the *Spag6l* gene may have acquired functions in addition to modulating cilia/flagella motility during evolution. Even though SPAG6 does not bind some proteins that bind to SPAG6L, both SPAG6 proteins bind TAC1, indicating that SPAG6 might have yet to be determined roles in vivo.

## 2. Materials and Methods

### 2.1. Ethics Statement

Guidelines of the Wayne State University Institutional Animal Care with the Program Advisory Committee (Protocol number: 21-01-3080) were observed in execution of all animal research.

### 2.2. Constructs for the Studies

The coding region of mouse *Spag6* cDNA was amplified by RT-PCR using the following primers: forward 5′-GAATTCATGAGCCAGCGGCAGGTGCTGCAA-3′; reverse 5′-GGATCCCGTTAATAAGAGGCTGATAGCTGTCG-3′. The PCR product was cloned into the pCR2.1 TOPO-TA vector. Fidelity of the PCR product was verified through sequencing and the cDNA was, subsequently, subcloned into EcoR1/BamHI sites of the pEGFP-N_2_ vector, pGBK-T7 vector, and pGAD-T7 vector. Mouse *Ccdc103* cDNA was amplified in a similar fashion using the following primers: forward 5′-GAATTCGCCATGGAGAAGAACGATGTAATC-3′; reverse 5′-GGATCCCATGGACTCCATACAGTTCTAGCAG-3′. Mouse *Ccdc103* cDNA was later subcloned into EcoR1/BamHI sites of the pGAD-T7 vector.

### 2.3. Animals and Genotyping

One pair of homozygous *Spag6* global knockout mice was purchased from The Jackson Laboratory (Bar Harbor, ME, USA; Stock# 033958), and the pair was used to generate homozygous mice for the study. Heterozygous mice obtained by breeding the homozygous male to a wild-type female were used as controls for the study. Genomic DNA was isolated from the toes of 7-day-old mice for characterization of genotype. Genotype was determined by PCR using the primer sets listed below: m*Spag6*F: 5′-GCATTTTCAGCACAGTTTGA-3′; m*Spag6*RWT: 5′-TGGGAACTGCCTGGGATATG-3′; m*Spag6*RMU: 5′-GGATTATAGGCATGTACCTTTGC-3′. The m*Spag6*F/m*Spag6*RWT pair amplified a wild-type band, and the m*Spag6*F/m*Spag6*RMU pair amplified a mutant band.

### 2.4. Histological Examination of Testicular and Epididymal Tissues

Testes and epididymides of adult mice were collected and fixed in 4% paraformaldehyde (PFA) in phosphate-buffered saline (PBS) at 4 °C overnight. The tissues were embedded in paraffin, sectioned at 5 μm thickness, deparaffined, and stained with hematoxylin and eosin, in accordance with standard procedures. Slides were examined using a BX51 Olympus microscope (Olympus Corp., Center Valley, PA, USA), and photographs were taken with a ProgRes C14 camera (JENOPTIK Laser, Jena, Germany).

### 2.5. RT-PCR

TRIzol (Invitrogen, Waltham, MA, USA) was utilized for total RNA extraction from mouse testes. cDNA was synthesized through reverse transcription using first-strand cDNA SensiFAST^TM^ cDNA Synthesis Kit (Meridian Bioscience, Cincinnati, OH). The cDNA was used as a template for RT-PCR using the following specific primers: (1) forward 5′-ATGAGCCAGCGGCAGGTGCTGCAA-3′; (2) reverse 5′-CGTTAATAAGAGGCTGATAGCTGT-3′; (3) reverse 5′GAGGAGAGGAGTGTTTACCAACCGC-3′; (4) forward 5′-GCGGTTGGTAAACACTCTCCTC-3′. Mouse *Gapdh* was amplified as the positive control using the following primers: m-*Gapdh* 452 forward 5′-TAACCTCAGATCAGGGCGGA-3′; m-*Gapdh* 452 reverse 5′-TGTAGGCCAGGTGATGCAAG-3’; a 452 bp product was amplified in all mice. To further confirm that the exon 4 was deleted in the *Spag6* KO mice, the PCR products amplified by P1/P2 primer pair from a KO mouse and a wild-type mouse were sequenced using the P1 primer, and the sequence results were compared to the *Spag6* cDNA sequence from GenBank (NM_001001334).

### 2.6. Western Blotting

Homogenization in radioimmunoprecipitation assay (RIPA) buffer was performed to mechanically lyse mouse testicular samples or cultured cells. Collected lysates were denatured by incubation at 95 °C for 10 min. The samples were, subsequently, loaded onto 12% sodium dodecyl sulfate-polyacrylamide gels, separated electrophoretically and transferred to polyvinylidene difluoride membranes (Millipore Sigma, Burlington, MA, USA). The membranes were blocked with Tris-buffered saline solution containing 5% non-fat dry milk and 0.05% TWEEN 20 for 1 h, followed by incubation with the indicated antibodies at 4 °C overnight. After washing in TBST, the blots were incubated with secondary antibodies conjugated with horseradish peroxidase for 1 h at room temperature. Following further washes, the proteins were detected with Super Signal chemiluminescent substrate (Thermo Scientific, Waltham, MA, USA). Antibodies used were: anti-GFP (Invitrogen, Waltham, MA, USA, 1:2000, Cat no: MA5-15256); anti-β-actin (Cell Signaling Technology, Danvers, MA, USA, 1:2000, Cat no: 4967S); anti-SPAG16L (1:1000, generated by our own laboratory); anti-COPS5 (Sigma-Aldrich, St. Louis, MO, USA, 1:2000, Cat no: J3020).

### 2.7. Cell Culture and Transient Transfection

COS-1 cells and Chinese hamster ovary (CHO) cells were cultured in DMEM (Invitrogen, Waltham, MA, USA) supplemented with 10% fetal bovine serum and 5% L-glutamine at 37 °C. Transfection was performed with Lipofectamine™2000 transfection reagent (Invitrogen, Waltham, MA, USA) in accordance with the company’s protocol. The cells were, subsequently, processed for Western blotting and immunofluorescence analyses. 

### 2.8. Immunofluorescence Staining of Cultured Mammalian Cells

CHO cells were cultured in chambered slides. After transfection, the cells were fixed in 4% paraformaldehyde/PBS at room temperature for 30 min, then washed with PBS. The cells were permeabilized with 0.1% Triton X-100 (Sigma-Aldrich, St. Louis, MO, USA) at 37 °C for 10 min and blocked with 10 % goat serum (in PBS) for 1 h. The cells were washed with PBS again and incubated with the indicated antibody (anti-α-tubulin: Proteintech, Rosemont, IL, USA, 1:300, Cat no: 11224-1-AP) at 4 °C overnight. After washing with PBS, the samples were incubated with Cy3-conjugated secondary antibody at room temperature for 1 h. The slides were washed with PBS and mounted in VectaMount with DAPI (Vector Labs., Burlingame, CA, USA) and sealed with nail polish. Images were taken by confocal laser-scanning microscopy (Leica SD600, Leica Microsystems, Wetzlar, Germany) and processed using Adobe Photoshop 5.0 (Adobe Systems, San Jose, CA, USA).

### 2.9. Male Fertility Test

Two-month-old *Spag6* KO and control mice were independently mated with two-month-old wild-type mice for one month. The presence of vaginal plugs was noted to validate the occurrence of mating, and pregnancy in females was recorded. The number of pups delivered in each litter was noted the day after birth. Average litter sizes were obtained using the following formula: $\frac{total\;number\;of\;pups\;born}{number\;of\;mating\;cages}$.

### 2.10. Sperm Parameters

After breeding analyses, carbon dioxide inhalation followed by cervical dislocation were performed to euthanize the mice. Sperm were collected from the cauda epididymis in 37 °C PBS solution. Sperm motility was observed using an inverted microscope (Nikon, Tokyo, Japan) equipped with 10× objective. Movies were recorded at 15 frames/sec with a SANYO (Osaka, Japan) color charge-coupled device, high-resolution camera (VCC-3972) and Pinnacle Studio HD (version 14.0) software (Corel, Ottawa, ON, Canada). In total, 10 fields were analyzed per sperm sample. Individual spermatozoa were tracked using Image J (National Institutes of Health, Bethesda, MD, USA) and the plug-in tool MTrackJ. Sperm motility was calculated as curvilinear velocity (VCL), which is equivalent to the curvilinear distance (DCL) traveled by each individual spermatozoon in 1 s (VCL = DCL/*t*). To quantify sperm number, fixation was first conducted by incubation of sperm with 4% paraformaldehyde for 15 min at room temperature. Cells were counted using a hemocytometer chamber under a light microscope, and total sperm number was extrapolated based on standard methods.

### 2.11. Direct Yeast Two-Hybrid Assay

SPAG6L/pGBK-T7, COPS5/pGAD-T7, and TAC-1/pGAD-T7 plasmids were constructed previously [10]. To conduct direct yeast two-hybrid assays, the yeast was transformed with the indicated plasmids using the Matchmaker™ Yeast Transformation System 2 (Clontech, Mountain View, CA, USA, Cat#: 630439). Two plasmids containing simian virus (SV), 40 large T antigen in pGAD-T7, and p53 in pGBK-T7 were co-transformed into AH109 yeast as a positive control.

### 2.12. Structural Analysis and Comparison

The structures of SPAG6 and SPAG6L were predicted using the AlphaFold2 algorithm [30] as implemented in ColabFold [31]. Five structures were generated for each (without relaxing the coordinates) and the lowest rank was compared. Alignment of the structures was performed using the McLachlan algorithm [32] as implemented in the program ProFit (Martin, A.C.R. and Porter, C.T., http://www.bioinf.org.uk/software/profit/. Accessed on 10 December 2021). Structure figures were generated using PyMOL (The PyMOL Molecular Graphics System, Version 2.0, Schrödinger, LLC, New York, NY, USA.).

## 3. Results

### 3.1. Spag6 Shares High Sequence Homology with Spag6l

Mouse *sperm-associated antigen 6* (NCBI reference sequence: NM_001001334.2) is 1651 bp long and consists of 11 exons. Mouse *sperm-associated antigen 6-like* (NCBI reference sequence: NM_015773.2) is 2483 bp long and consists of 12 exons, the last of which is a non-coding exon. A comparison of the coding regions of *Spag6* and *Spag6l* revealed a high sequence homology (Appendix A). The main difference that distinguishes the two genes lies within exon 11. *Spag6* contains six additional nucleotides prior to the stop codon that are absent from *Spag6l*. The non-coding regions of exon 1 and 12 in *Spag6l* also contain additional nucleotides that are not present in *Spag6* (Figure 1A, Appendix A). These differences in the nucleotide sequences allowed us to design primers to amplify the two *Spag6* cDNAs separately. The two proteins are also conserved. They share 93% identity in amino acid composition (Figure 1B).

### 3.2. Spag6 and Spag6l Have Different mRNA Distributions In Vivo

To compare the mRNA distributions of the two *Spag6* genes, specific primers that target *Spag6s* were synthesized and RT-PCR was conducted. *Spag6l* mRNA was detected in all of the tissues examined (Figure 2A). However, *Spag6* mRNA was only detected in brain and the testis, which had the highest level of expression. No *Spag6* mRNA was detected in the spleen, kidneys, or lungs (Figure 2B).

### 3.3. SPAG6 Protein Has a Similar Localization as SPAG6L in Transfected Mammalian Cells

To evaluate the localization of SPAG6 in transfected mammalian cells, the SPAG6/pEGFP-N_2_ plasmid was constructed. To test SPAG6/GFP fusion protein expression, COS-1 cells were transfected with an empty pEGFP-N_2_ plasmid and the SPAG6/pEGFP-N_2_ plasmid. A Western blot analysis using the specific anti-GFP antibody revealed that the SPAG6/GFP fusion protein was expressed in the transfected COS-1 cells (Figure 3A). The plasmid was then transfected into CHO cells, and the localization of the SPAG6/GFP fusion protein was examined. In these cells, SPAG6 had a similar localization as SPAG6L (Appendix A). To determine if SPAG6 decorated microtubules similar to SPAG6L (3, 7), the transfected cells were stained with an anti-α-tubulin antibody. Similar to SPAG6L/GFP, SPAG6/GFP also colocalized with a subset of microtubules (Figure 3B).

### 3.4. Mice Deficient in Spag6 Gene Expression Are Grossly Normal and Fertile

A mouse knockout model was generated by The Jackson Laboratory using the CRISPR/cas9 system to disrupt the *Spag6* gene on chromosome 2. Exon 4 of the gene was targeted for deletion (Figure 4A). To validate the mutation, RT-PCR was conducted using testicular cDNA with different primer sets. Given that exon 4 was deleted in the *Spag6* KO mice, a 1.4kb PCR product was amplified from the KO mice when primer set P1/P2 was used, while a 1.6 kb full-length *Spag6* cDNA was amplified from the wild-type mice (Figure 4B(a)). Primer P3 is a reverse primer located in the deleted exon of the *Spag6* gene as well as in exon 4 of the *Spag6l* gene. When primer set P1/P3 was used, the 346 bp PCR products were not only amplified from the wild-type mice, but also from the *Spag6* KO mice (Figure 4B(b)). P4 is a forward primer localized in the deleted exon 4, and P2 is a reverse primer in exon 11 only targeting *Spag6*, not *Spag6l*. When the primer set P4/P2 was used, the 1.2 kb PCR product was only amplified from the wild-type mice, not from the *Spag6* KO mice (Figure 4B(c)). To further confirm that exon 4 was deleted in the *Spag6* KO mice, the PCR products amplified by the P1/P2 primer set from a KO mouse and a wild-type mouse were sequenced using the P1 primer, and the sequence results were compared to the mouse *Spag6* mRNA sequence in GenBank. The wild-type mouse had an intact exon 4 sequence (Appendix A); however, exon 4 was missing in the KO mouse (Appendix A). An in vitro translation assay revealed that an early stop code was created after exon 4 was deleted (Appendix A).

All *Spag6* KO mice were grossly normal (Appendix A), and there was no significant difference in body weight between the heterozygous control and homozygous knockout mice (Appendix A). Hydrocephalus, which was observed in the *Spag6l* knockout mice [4], was not seen in any of the *Spag6* KO mice (Appendix A). No abnormal behavior was observed in any of the control and homozygous mice analyzed throughout the studies.

Fertility of the homozygous *Spag6* mutant mice was examined. All female and male mice examined were fertile, with a comparable number of pups delivered in each litter (Table 1).

### 3.5. Spag6 KO Mice Have Normal Sperm Parameters

Epididymal sperm from the control and *Spag6* KO mice were examined. Sperm from the two genotypes appeared to exhibit normal morphology with hook-shaped heads and smooth, elongated tails (Appendix A, Figure 5A,B). Sperm numbers in the KO mice were not significantly different from those in controls (Figure 5C). Sperm motility was also examined. Sperm from both control and KO mice were mostly motile and exhibited progressive forward movement (Appendix A, Figure 5D,E).

### 3.6. Spermatogenesis Is Not Affected in Spag6 KO Mice

To examine spermatogenesis in *Spag6* KO mice, testes from 3–4-month-old heterozygous and *Spag6* KO mice were collected and tissue sections were stained with H&E. The testis size appeared to be normal in the *Spag6* KO mice, and there was no difference in testis weight/body weight between the control and *Spag6* KO mice (Appendix A). Light microscopy revealed that in heterozygous and homozygous adult mice, the seminiferous tubules showed a normal structure. Spermatids were well arranged with long tails extending into the lumen. Normal spermiation was observed (Figure 6A, Appendix A). The histology of epididymides was also examined. In the heterozygous mice, typical adult sperm contents were found (Figure 6B, upper). A similar sperm content was observed in the epididymides from homozygous adult mice (Figure 6B, lower). The lumen of cauda epididymis from both genotypes had compacted sperm with well-aligned heads and tails. Sloughed round spermatids were rarely observed in both homozygous and heterozygous mutant mice. 

### 3.7. Testicular Expression of Selected SPAG6L-Binding Partners in the Spag6-Deficient Mice

We previously discovered that expression levels of some SPAG6L-binding partners were dramatically reduced in the *Spag6l* KO mice [10]. We evaluated expression levels of selected proteins in the *Spag6* KO mice. SPAG16L was found to be dependent on the expression of SPAG6L. The Western blot analysis revealed that there was no difference in expression levels of SPAG16L and COPS5, another SPAG6L binding partner, between the control and *Spag6*-deficient mice (Figure 7).

### 3.8. Structural Differences between SPAG6 and SPAG6L

The predicted structures of both SPAG6 and SPAG6L had nearly identical overall folds (Figure 8A) consisting of eight armadillo repeats with an overall backbone (Cα) root mean square deviation of 0.730 Å (residues 1-507). The amino acids that differed between the two proteins are shown as spheres and colored yellow (Figure 8B), which shows that they were spread throughout the domain and largely solvent-exposed. A few of these amino acid differences formed small clusters on the protein surface (circled in Figure 8B), which suggests possible binding sites that differentially interacted with target proteins.

### 3.9. SPAG6 and SPAG6L Bind to Different Proteins

Both SPAG6 and SPAG6L have eight contiguous armadillo domains that mediate protein–protein interaction. The differences in structure of the two proteins suggests that they have different abilities to bind to other proteins. We identified a number of binding partners of SPAG6L in a yeast two-hybrid screen [10]. We decided to select two of the SPAG6L binding partners, TAC1 and COPS5, and tested if SPAG6 also bound to these two proteins. Direct yeast two-hybrid assays revealed that both SPAG6 and SPAG6L bound to TAC1 (Figure 9, left). However, COPS5 interacted with SPAG6L but not SPAG6 in yeast (Figure 9, right). 

## 4. Discussion

Here, we compared two mouse *Spag6* genes, the ancient *Spag6* gene that is located on chromosome 2, and the evolved one, *Spag6l*, which is located on chromosome 16. Even though the two *Spag6* genes shared high similarities in their nucleotide and amino acid sequences, they had different expression patterns *in vivo*. The ancient *Spag6* gene was expressed in a limited number of tissues, including the testis and brain. In contrast, the *Spag6l* gene was active in all tissues examined, including tissues without motile cilia, such as the spleen and kidneys. The expression of *Spag6l* in tissues without motile cilia suggests that the evolved gene acquired additional functions during evolution, which was supported by the unexpected phenotypes identified in the *Spag6l* KO mice, including hearing loss, impaired spermatogenesis, and immunological synapse formation [5,6,7,8,9,10,11]. It should be noted that humans only have one known *SPAG6* gene, and it remains to be determined if other species have multiple expressed *Spag6*-related genes.

Different expression patterns in vivo suggested that the two *Spag6* genes have different regulatory mechanisms. Even though the coding sequences were highly similar, *Spag6l* had a longer 5′-UTR in exon 1. In addition to the 11 exons present in both *Spag6* genes, *Spag6l* had an additional non-translated exon 12. Gene regulation through UTRs is one of the key mechanisms that governs gene expression [33]. Transcription factors/RNA binding proteins might bind to these UTRs to regulate *Spag6l* expression. Thus, the unique non-translated sequences in the *Spag6l* gene could contribute to its different expression pattern compared to the *Spag*6 gene in vivo.

Given that the two SPAG6 proteins were highly similar in their amino acid sequences, it is not surprising that they had a similar localization in transfected CHO cells. Both proteins decorated a subset of microtubules. Our earlier studies suggested that acetylated α-tubulin might account for the subset of microtubules (7). However, it is not clear if the two proteins have the same subcellular localization in vivo. We generated an anti-SPAG6L antibody in rabbits using purified full-length SPAG6L [3]. The antibody should have cross-reacted with both SPAG6 proteins. Differences in some amino acids between the two SPAG6 protein sequences raised the possibility of generating antibodies that recognize the individual SPAG6 proteins. However, none of the six antibodies we created using synthetic peptides were able to distinguish the two proteins. Thus, specific antibodies against the two SPAG6 proteins are still needed to characterize the localization of the two SPAG6 proteins in vivo.

Even though no specific antibody targeting SPAG6 is available, RT-PCR using specific primers demonstrated that the *Spag6* gene on chromosome 2 was specifically disrupted in our knockout model, and that *Spag6l* was not affected. However, the phenotype of the *Spag6* KO mice was totally different from the *Spag6l* KO. The phenotypes reported in the *Spag6l* KO mice, including hydrocephalus, impaired sperm motility, and abnormal spermatogenesis were not discovered in the *Spag6* KO mice. As an ancient gene, the translated SPAG6 protein may only retain its ancient functions. However, in addition to the ancient functions, SPAG6L protein appears to have acquired other functions during evolution. Therefore, it is likely that SPAG6L compensates for the function of SPAG6 when *Spag6* is disrupted. Conversely, SPAG6 may only partially compensate for the function of SPAG6L when the *Spag6l* gene is disrupted. We expect that *Spag6/Spag6l* double KO mice will have a more severely affected phenotype.

The different functions of the two SPAG6 proteins may be determined by the few different amino acids between the two proteins, even though both SPAG6 proteins contain armadillo repeats that mediate protein–protein interaction [34]. Given the high sequence identity, the predicted overall fold of the two proteins was nearly identical. Of note, though, a few of the amino acid differences appeared to form small clusters on the surface, which may represent differential binding sites. One example was the interaction with COPS5. The protein was identified to be a binding partner of SPAG6L [10]. However, SPAG6 did not interact with COPS5 in our direct yeast two-hybrid assay. However, the two SPAG6 proteins both bound to TAC1. Thus, it appears that during evolution, due to changes in a few solvent-exposed amino acids, SPAG6L acquired the ability to interact with proteins that do not bind to the ancient SPAG6, which may allow SPAG6L to perform new functions. It will be interesting in the future to test if SPAG6 binds to other SPAG6L binding partners identified in our previous study. Our Western blotting results examining selected SPAG6L binding partners, including SPAG16L, did not support SPAG6 binding to the protein. A yeast two-hybrid screen using SPAG6 as the bait may identify different binding partners than those that interact with SPAG6L. However, given the phenotype of the *Spag6* knockout mice, the gene is probably less important than *Spag6l* in tissues with motile cilia.

A change in binding partner profiles during evolution is supported by another recent study. In zebrafish, the SPAG6 protein, whose sequence is 80% identical to mouse SPAG6, associates with Ccdc103, a coiled-coil domain-containing protein related to primary ciliary dyskinesia [35]. However, in our direct yeast two-hybrid assay, neither SPAG6 nor SPAG6L interacted with Ccdc103 (Appendix A), suggesting that even though Ccdc103 and SPAG6 interact in zebrafish, they do not interact in mice. Thus, even though both mouse SPAG6 proteins have conserved armadillo-repeat domains that mediate protein–protein interaction, the few different amino acids might endow the two proteins with common, but also some different binding partners, accounting for the overlap in binding partners, but also differences.

In the present study, we only focused on the gross phenotype and reproduction of the *Spag6* KO mice. It remains to be determined if the ancient SPAG6 protein performed other functions in vivo, particularly roles in polarity, cell growth, ciliogenesis, hearing and immunologic functions as we identified in the *Spag6l* KO mice. Thus, further studies need to be conducted to fully elucidate the function of the ancient *Spag6* gene, including its impact on CNS function. The two *Spag6* mouse models provide tools to study the impact of evolution on these related genes.

## Figures and Tables

**Figure 1 cells-11-00336-f001:**
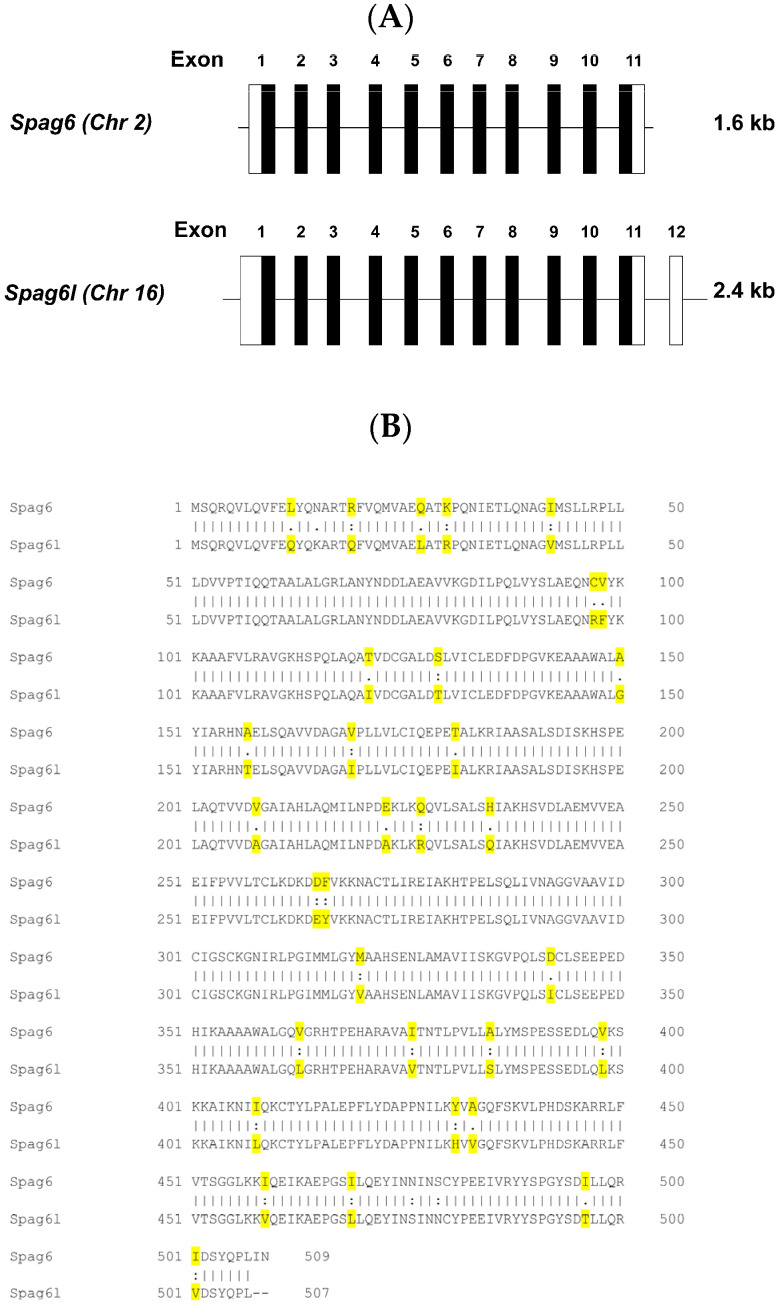
The ancient *Spag6* shares high homology with the evolved *Spag6l* gene. (**A**) Gene structures of mouse *Spag6* and *Spag6l*. Genomic structures of the *Spag6* and *Spag6l* genes and exons included in the 1.6 and 2.4-kb transcripts. The filled blocks represent the coding exons, and the open blocks represent the non-coding regions. (**B**) Amino acid homology between SPAG6 and SPAG6L. They share 93% identity in amino acid sequence. Different amino acids in the two protein sequences are highlighted in yellow. Colon ( : ) indicates conservation between groups of strongly similar properties—scoring > 0.5 in Gonnet PAM 250 matrix. Period ( . ) indicates conservation between groups of weakly similar properties—scoring < 0.5 in the Gonnet PAM 250 matrix.

**Figure 2 cells-11-00336-f002:**
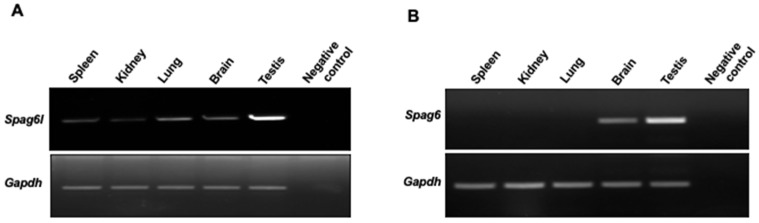
*Spag6* and *Spag6l* have different mRNA distributions in vivo. Total RNA was extracted from the indicated tissues, and RT-PCR was conducted using specific primer sets targeting *Spag6* and *Spag6l*. (**A**) RT-PCR to examine *Spag6l* mRNA expression; (**B**) RT-PCR to examine *Spag6* mRNA expression. Expression of *Gapdh* was examined as a control. Notice that *Spag6* mRNA was detected in the testis and brain, but not in other tissues examined. However, *Spag6l* mRNA was detected in all tissues examined.

**Figure 3 cells-11-00336-f003:**
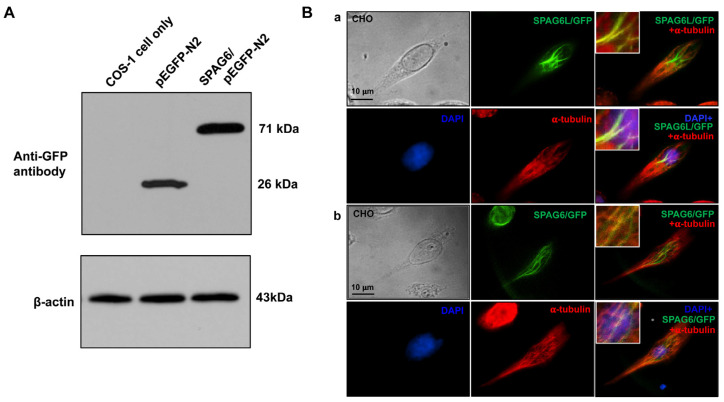
SPAG6 decorates a subset of microtubules in the transfected CHO cells. (**A**) Examination of SPAG6/GFP fusion protein expression in the transfected COS-1 cells by Western blot analysis. The COS-1 cells were transfected with an empty pEGFP-N_2_ plasmid or a SPAG6/pEGFP-N_2_ plasmid; 48 h after transfection, the cells were collected into RIPA buffer and Western blotting was conducted using an anti-GFP monoclonal antibody. (**B**) SPAG6/GFP colocalized to a subset of microtubules in the transfected CHO cells. The CHO cells transfected with SPAG6L/pEGFP-N_2_ (**a**) or SPAG6/pEGFP-N_2_ (**b**) were stained with an anti-α-tubulin antibody, and imaged using a Leica SD600, LDI-7 confocal microscope. Notice that both SPAG6/GFP and SPAG6L/GFP co-localized with a subset of microtubules.

**Figure 4 cells-11-00336-f004:**
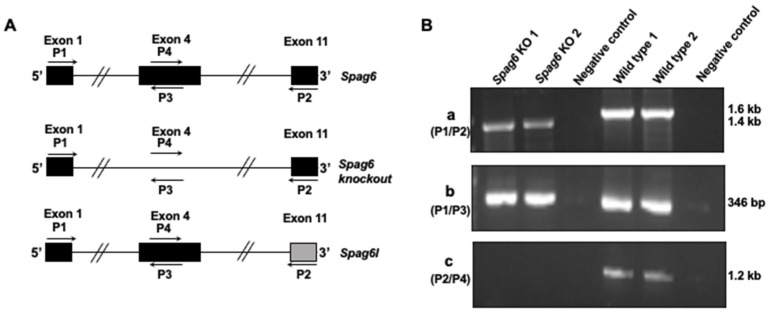
Generation of global *Spag6* KO mice. (**A**) Schematic representation of the strategy used to disrupt the *Spag6* gene on chromosome 2. The arrows indicate specific primers used to amplify different regions of *Spag6* cDNA. Solid bars are representative of the indicated exons. Notice that exon 11 of *Spag6l* is different from *Spag6*, and P2 does not bind to exon 11 of *Spag6l* cDNA; (**B**) Respective PCR results amplified by the indicated primer sets. (**a**): primers 1 and 2; (**b**): primers 1 and 3; (**c**): primers 4 and 2. A smaller PCR product was amplified from the *Spag6* knockout mice when P1/P2 primer set was used, indicating partial deletion of *Spag6* cDNA sequence. The PCR product amplified using P1/P3 primer set from the *Spag6* knockout mice is likely from the *Spag6l* cDNA; no PCR product was amplified using P4/P2 primer set from the *Spag6* knockout mice because P2 does not bind to *Spag6l* cDNA.

**Figure 5 cells-11-00336-f005:**
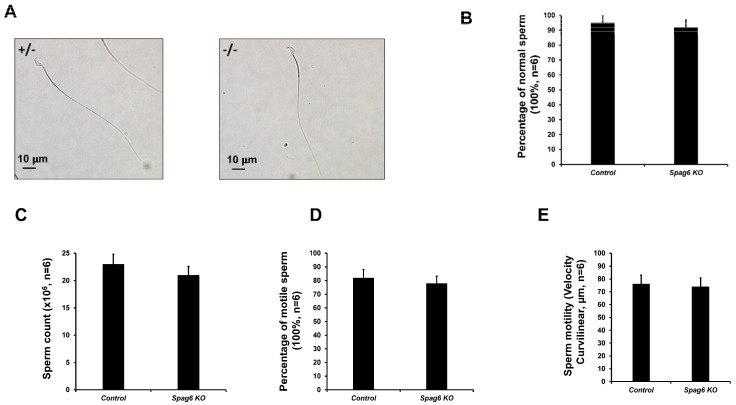
Normal sperm parameters in *Spag6* KO mice. (**A**) Representative images of epidydimal sperm of the control and *Spag6* knockout mice at high magnification. Notice that sperm look to be normal in both control and the *Spag6* knockout mice; (**B**) percentage of normal sperm morphology. There were no significant differences between the control and *Spag6* knockout mice. N = 4 for each group; (**C**) sperm number in the heterozygous and homozygous *Spag6*-deficient mice. There were no significant differences between the two groups. N = 4 for each group; (**D**) percentage of motile sperm in the heterozygous and homozygous *Spag6*-deficient mice. N = 4 for each group; (**E**) percentage of sperm motility in the heterozygous and homozygous *Spag6*-deficient mice. N = 4 for each group.

**Figure 6 cells-11-00336-f006:**
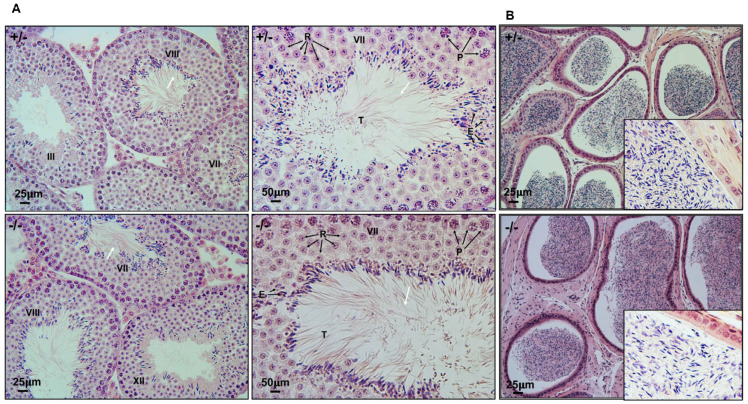
Normal spermatogenesis in the *Spag6* KO mice. (**A**) Representative H&E staining of the adult testis of a control (+/−) mouse and a *Spag6*-KO littermate. The Roman numerals represent the various stages of the tubules. The control testis showed normal seminiferous tubules structure, with normal spermiation and release of spermatozoa (white arrows). In *Spag6*-deficient testis, a similar histology was observed. P: pachytene spermatocytes; R: round spermatids; E: elongating spermatids; T: tail of sperm being released; (**B**) representative H&E staining of the adult cauda epididymis of a control (+/−) mouse and a *Spag6*-KO littermate. Notice that the lumens were filled with sperm in mice of both genotypes. The insets show high magnification of images.

**Figure 7 cells-11-00336-f007:**
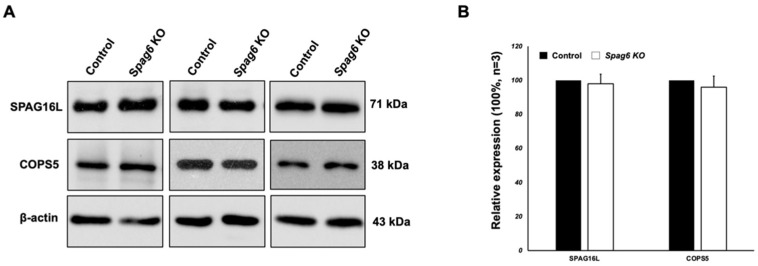
Expression of selected protein partners of SPAG6L was not affected by SPAG6 deficiency. Examination of testicular SPAG16L and COPS5 protein levels in the *Spag6* KO mice by Western blot analysis. Notice that there was no difference in their levels between the control and KO mice. (**A**) Western blot results; (**B**) relative levels of SPAG16L and COPS5 normalized to β-actin.

**Figure 8 cells-11-00336-f008:**
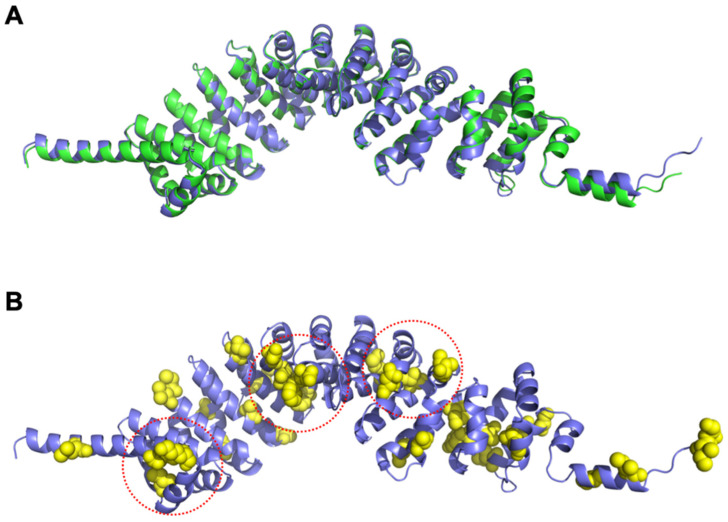
SPAG6 and SPAG6L have distinct protein conformations. (**A**) Ribbon diagrams of the aligned SPAG6 (blue) and SPAG6L (green) predicted structure show nearly identical folds for the two proteins consisting of eight armadillo repeats; (**B**) the sidechains for amino acids that differed between SPAG6 and SPAG6L are shown in yellow spheres on the SPAG6 structure with three clusters of these differences circled in red.

**Figure 9 cells-11-00336-f009:**
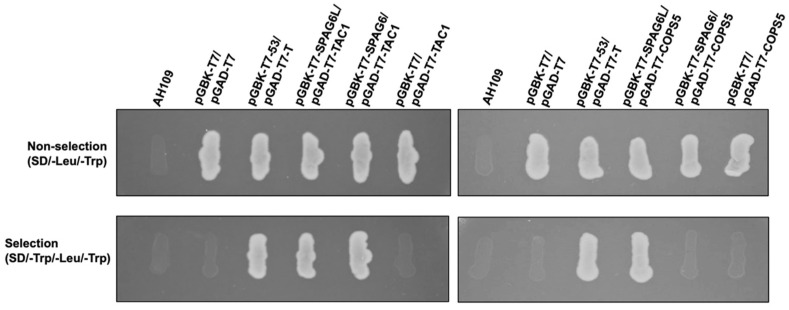
SPAG6 and SPAG6L have different binding partners. Direct yeast two-hybrid assay to analyze interaction between the two SPAG6 proteins and TAC-1 (**left**) and COPS5 (**right**). Indicated plasmids were transformed into AH109 yeast. The transformed yeast grew on the plates with non-selection medium (SD-2) and selection medium (SD-3). Pair of P53/LgT was used as the positive control. Notice that both SPAG6 proteins interacted with TAC-1 in the assay; however, COPS5 only interacted with SPAG6L. The images are representative of three independent experiments.

**Table 1 cells-11-00336-t001:** Fertility and fecundity of the *Spag6* KO mice. To test fertility, 8 weeks old males were bred with wild-type fertile females for one month, and litter size was recorded for each mating. ^a^ Number of fertile mice/total number of mice.

*Spag6* Genotype	Male Fertility ^a^	Litter Size	Female Fertility ^a^	Litter Size
+/−	6/6	7.9 ± 0.5	6/6	8.1 ± 0.6
−/−	6/6	7.6 ± 0.7	6/6	7.4 ± 0.5

## Data Availability

Not applicable.

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
