# Peer review of "The Ancient and Evolved Mouse Sperm-Associated Antigen 6 Genes Have Different Biologic Functions In Vivo"

_cells, 2022, doi:10.3390/cells11030336_

Round 1

Reviewer 1 Report

This manuscript describes the role of the ancient sperm-associated protein antigen 6 (SPAG6) protein in male fertility. Like the homologous SPAG6L, which shares 93% identity to SPAG6, the authors found that SPAG6 localizes to microtubules in cultured cells. Although SPAG6L-KO mice show multiple intriguing phenotypes including male infertility, unexpectedly, SPAG6-KO mice are grossly normal and show normal fertility in both males and females. The authors provide some evidence that their distinct functional characteristics may be attributable to differences in solvent exposed amino acids and binding partners.

I agree that this is an interesting report suggesting that the ancestral mouse spag6  gene may have lost its key functions during evolution and should be published.  However, there are several issues that must be resolved prior to acceptance.

Major critiques:

  1. The reason that the authors claim that spag6 is ancient is not clear. I assume it is based on evolutionary conservation. It would be helpful to support this if a phylogenetic tree is provided.

  1. There is no solid evidence that the authors created a complete spag6-knockout mouse line. I understand that generating specific antibodies is difficult, but western blotting using pan SPAG6 antibodies may show a 50% reduction in SPAG6/SPAG6L protein levels. Alternatively, authors could sequence the spag6 cDNA (1.4 kb, Fig4B) from SPAG6-KO mice to make sure that the coding region contains a frameshift that leads to a truncation in its N-terminal region.

  1. It is not clear if SPAG6 and SPAG6L localize to microtubules (Fig3B and S2). Please provide high mag images or show costainig with tubulins. Similarly, in the phase-contrast images, cells are barely seen. Please replace these images.

  1. Sperm images in FigS4 are of low quality. Please provide better images so that readers can assess sperm morphology.

  1. Fig8B is difficult to interpret. Does each yellow sphere represent an amino acid? Perhaps, different aa can be highlighted in the protein alignment in Fig1B.

Minor critiques:

  1. There are MANY typos and grammatical errors noted throughout the text. Please correct. Just to list a few examples:

1) p2, Spag6 gene was renamed as Spag6 should be Spag6l gene was renamed as Spag6.

2) p3, primers specific should be specific primers.

3) p3, ccDc103 should be Ccdc103.

4) p5, (Supplemental Figure 1A) should be (Supplemental Figure 1).

5) p5, (Fig. 1A, Supplemental Figure 1A) should be (Fig. 1A).

6) p7, a fluorescence microscopy should be a fluorescence microscope.

7) p8, chromosome 16 should be chromosome 2.

  1. For FigS3B, Ns are missing (numbers of mice used to calculate the body weights).

  1. In the methods, it says that fertility tests were done for one month, but it is not consistent at multiple locations in the text.

  1. p12, “As an ancient gene, the translated SPAG6 protein may only retain its ancient function: to modulate cilia/flagella motility.” might be an overstatement.

Author Response

Please see the attachment for point-by-point response to the reviewer's comments. Thank you. 

Reviewer 2 Report

In the proposed article, a comparative study of two mouse genes ancient Spag6 and evolved Spag6l were compared. The findings are interesting and scientifically important. To my great regret, the morphological part of the study does not correspond in its level to the molecular biological part of the work. The article is of interest and may, after revision, be accepted for publication in the journal Cells. 

Major points:
Page 6 Figure 3. 
What is your statement about colocalization of the SPAG6 protein with microtubules based on? For this approval, you must do a double immunofluorescence stain for SPAG6 protein and microtubules. The quality of immunofluorescent illustrations (Fig. 3B, Supplemental Figure 2) in the proposed form is also not good enough for publication in the journal Cells (from my own experience I know that the quality of photographs sometimes deteriorates greatly when converted to PDF format, but microtubules are not visible in the photographs - only diffuse green labelling). It is also necessary to complement these illustrations with insertion photos of individual cells at high magnification, taken using x63 or x100 immersion objectives with a good aperture, so that fine details of the labelling can be seen.

Page 9 Figure 6.
Histological photographs are also of insufficient quality. Again, it is necessary to complement these illustrations with inset photographs of small regions at high magnification, taken using x63 or x100 immersion objectives with a good aperture, so that we can see the fine details of the structure of the testicles. 

I did not see the white arrows, about which it is written in the legend to the figure.

Minor points:
It seems Abstract is too long for Cells journal paper?

Page 3, 2 line from down 
In the text:    …containing 5% non-fat dry and 0.05% Tween 20)
Must be :         …containing 5% non-fat dry milk and 0.05% Tween 20)

Page 10 Figure 7.
ciency.Examination
No space between point and letter   

Discussion line 4  
Even though the two Spag6 genes share high similarities in their nucleotide and amino acid sequences, they have different expression patterns in vivo.
May be amino acid sequences of proteins? 

Page 12, 10 line from down 
ary dyskinesia and ??? [35]. However
Missing word?

Author Response

Please see the attachment for point-by-point response to the reviewer's comments. Thank you so much. 

Round 2

Reviewer 2 Report

Dear Editor!
The authors took into account all my comments and presented high quality illustrations. In this form, the paper can accepted for publication.